# Trans Species RNA Activity: Sperm RNA of the Father of an Autistic Child Programs Glial Cells and Behavioral Disorders in Mice

**DOI:** 10.3390/biom14020201

**Published:** 2024-02-07

**Authors:** Zeynep Yilmaz Sukranli, Keziban Korkmaz Bayram, Ecmel Mehmetbeyoglu, Zuleyha Doganyigit, Feyzullah Beyaz, Elif Funda Sener, Serpil Taheri, Yusuf Ozkul, Minoo Rassoulzadegan

**Affiliations:** 1Betul-Ziya Eren Genome and Stem Cell Center, Erciyes University, Kayseri 38039, Turkey; 2Department of Medical Genetics, Faculty of Medicine, Erciyes University, Kayseri 38039, Turkey; 3Department of Medical Genetics, Faculty of Medicine, Yıldırım Beyazıt University, Ankara 06010, Turkey; 4Histology and Embryology Department, Medical Faculty, Yozgat Bozok University, Yozgat 66700, Turkey; 5Histology and Embryology Department, Faculty of Veterinary, Erciyes University, Kayseri 38039, Turkey; 6Department of Medical Biology, Faculty of Medicine, Erciyes University, Kayseri 38039, Turkey; 7The National Institute of Health and Medical Research (INSERM)-Centre National de la Recherche Scientifique (CNRS), Université Côte d’Azur, Inserm, 06000 Nice, France

**Keywords:** non-Mendelian, autism, heredity, sperm, microRNAs, transcription, mouse model, RNA microinjection, glial cells, astrocyte, microglia, astrogliosis, microgliosis

## Abstract

Recently, we described the alteration of six miRNAs in the serum of autistic children, their fathers, mothers, siblings, and in the sperm of autistic mouse models. Studies in model organisms suggest that noncoding RNAs participate in transcriptional modulation pathways. Using mice, approaches to alter the amount of RNA in fertilized eggs enable in vivo intervention at an early stage of development. Noncoding RNAs are very numerous in spermatozoa. Our study addresses a fundamental question: can the transfer of RNA content from sperm to eggs result in changes in phenotypic traits, such as autism? To explore this, we used sperm RNA from a normal father but with autistic children to create mouse models for autism. Here, we induced, in a single step by microinjecting sperm RNA into fertilized mouse eggs, a transcriptional alteration with the transformation in adults of glial cells into cells affected by astrogliosis and microgliosis developing deficiency disorders of the ‘autism-like’ type in mice born following these manipulations. Human sperm RNA alters gene expression in mice, and validates the possibility of non-Mendelian inheritance in autism.

## 1. Introduction

Autism is one of the most common behavioral changes detected early in life [1]. Despite extensive research, the etiology of autism remains very complex. The cause of autism is linked to the genetic and environmental history of parents, which together could lead to the development of autism spectrum disorders (ASDs) in children [1,2]. A recent survey reveals several hundred distinct sites of nucleotide changes in the genome (coding and essentially noncoding regions) between patients [3]. Several environmental and genetic factors identified in humans have already been validated with mouse models [4,5,6], but, have not yet been associated in all autistic patients [7,8,9]. Recently, we reported that the altered levels of six miRNAs (miR-19a-3p, miR-361-5p, miR-3613-3p, miR-150-5p, miR-126-3p, miR-499a-5p) in parents, and the transmission of an even more reduced expression compared to controls of the same miRNAs and now with altered phenotypes in patients [10], therefore, lead to the inclusion of a genetic marker common to cases of autism. Altered levels of RNAs such as miRNAs/noncoding RNAs can be passed on to the next generation via the germline to alter offspring phenotypes [11,12,13]—see the recent review on the broad regulatory network of miRNAs [14]. Thus, non-Mendelian inheritance with transcriptional alteration could play an important role in the etiology of autism.

However, how RNA subornation is established and maintained throughout generations remains a matter of speculation. Recently, on the activity of sperm RNA characterized in mouse models, a growing number of experiments highlight it as an important source of non-Mendelian hereditary information above DNA [15,16]. The six miRNAs listed above are present in sperm RNA in general and are altered in the father (normal) of an autistic child. Five of the six mRNAs (miR-19a-3p, miR-150-5p, miR-126-3p, miR-361-5p, miR-499a-5p) are also present in mouse sperm, and the sixth miR-3613-3p does not exist in the mouse genome. These altered hereditary miRNA profiles are then found downregulated in the sera of autistic patients and ‘autism-like’ mouse models. One of the questions posed here is to examine sperm RNA activities that cover a possible paternally acquired autistic phenotype. Microinjection of sperm RNA into fertilized mouse eggs could lead to variable phenotypes depending on the origin of the males from which the RNA is collected [17]. In particular, although exposure of total RNA from sperm (human) to fertilized mouse eggs may have adverse effects on offspring, can symptoms of autism also become established in the next generation? Here, we focus on experimentally induced autistic traits, particularly the potential of human sperm RNAs to induce “autism-like” phenotypes in mice.

The offspring of mice born from eggs microinjected with sperm RNAs from the father of the autistic child were compared to controls at the molecular level (expression of miRNAs in blood and hippocampus) and through behavioral assessment (open field maze, recognition of a familiar and novel object, social interaction, tail suspension, marble burying, and elevated plus maze assays).

In addition, we examined the prefrontal cortex, corpus callosum, striatum, hippocampus, amygdala, and cerebellum, which have been implicated in the pathophysiology of autism in the brain [18,19]. Our study involved, the examination of the brain anatomy in animals from all groups, utilizing immunohistochemical stains. Notably, differences were observed in animals induced by valproic acid (VPA) and microinjected RNA. While astrocytes showed abnormal astrogliosis (and microgliosis, which is normally invaded by microglia), no signs of astrocyte disorientation or microglial extensions, which are often seen in the autistic human brain, were detected. Moreover, we did not observe an increase in size (hypertrophy) in GFAP and Iba-1-positive cells, nor observe an increase in the number of immunopositive cells (hyperplasia).

Microglia and astrocytes play important roles in the regulation of neuronal metabolism, providing nutritional support to neurons, participating in synapse formation, and facilitating neurotransmission [20]. Among the alterations induced in glial cells following RNA microinjection, we observed a significant increase in the number of branches in oligodendrocytes, compared to the control groups. The results of glial cell alterations evidenced here are entirely consistent with other observations in the field, attributing a role to glial cells in brain development and their impairments in neuronal development disorders (NDDs) [21] and with autistic traits observed in mice.

Here, we found that groups of mice born after microinjection of RNA from the sperm of the father of autistic children, or of RNA from the sperm of adult mice treated with valproic acid at the age of two weeks, and candidate miRNAs (miR-499-3p) manifest autistic symptoms in the form of behavioral disorders with impairment linked to glial cell disorders.

## 2. Methods

### 2.1. Patient Selection Criteria

This study was approved by the Hospital Ethics Committee of Erciyes University School of Medicine. A detailed description of the study was given to all participants and their parents before their enrollment. All parents gave written informed consent before participation. The diagnosis was made by a multidisciplinary team (comprising an experienced child psychiatrist, a pediatric neurologist, and a genetic specialist), according to the criteria of the Diagnostic and Statistical Manual, Fourth and Fifth Edition, Text Revision (DSM-IV-TR; American Psychiatric Association, 2000 and DSM-V; American Psychiatric Association, 2013) criteria, using childhood autism rating scale (CARS). From the Turkish cohort of 37 families including one or more children with behavior disorders (45 subjects altogether) [10], we were interested in a family with two affected children. A father from two separate marriages has several children and each mother gave birth to a child who developed a different psychological disorder (see Appendix A). We have previously shown that all of them (six miRNAs) are already altered in their father’s sperm. For this reason, we planned to microinject RNA from the children’s father’s sperm into mouse embryos. For control, sperm RNA from the healthy control father was collected.

### 2.2. Human Sperm RNA Preparation

The same microRNAs that were reduced in the blood of autistic patients were also altered in the sperm RNAs of the father of autistic children [10]. To follow any effect from the human sperm RNAs in mice, we prepared total sperm RNAs from two men, one control and the second from the father of the two autistic children. Total sperm RNAs were prepared via standard Trizol extraction techniques (see Section 2). RNAs for microinjection into fertilized mouse eggs were adjusted to 0.2 ng/microliter from control sperm and sperm of the father of autistic children (see Appendix A for the number of animals and concentrations of RNAs and microRNAs). Mice born from microinjection were designed SH* (sperm human SHA* for autism), SH (sperm human for control), control (non-microinjected), and miR* (miRNA).

### 2.3. Mouse Husbandry

Mice were maintained according to the European regulations for the care and use of research animals. The genetic backgrounds are *B6D2* F1 and *Balb/c*.

### 2.4. RNA Microinjection

Oligoribonucleotides synthetic miRNA were adjusted to a concentration of 1 µg/ml, of which 1–2 pl (picoliter) was microinjected into normal Balb/c fertilized eggs according to established transgenesis methods [11]. Oligoribonucleotides were obtained from Eurofine (sequences provided in Appendix A).

Briefly, for collecting embryos, female and male mice were mated at 3 p.m. The day after, mated females were checked for the vaginal plug at 8 a.m. and vaginal plug (+) females mice were selected. Females were sacrificed that afternoon, and their oviducts were enclosed in the M2 medium (Sigma, Darmstadt, Germany). Then, embryos were collected under a bino-microscope (Leica, Wetzlar, Germany) and were transferred into M16 medium (Sigma, Germany) with a small thin glass transfer pipette (maneuvered by mouth) and incubated at 37 °C in 0.5% CO_2_ (Panasonic, Osaka, Japan). To capture the pronuclei at the most prominent stage (at 2–6 p.m.), we microinjected 1–2 pl of solution of oligonucleotides 1–5 ng/µL (see in Appendix A) with a glass pipette into the male pronucleus of embryos under an inverted microscope (Nikon, Melville, NY, USA). After the microinjection, the embryos were incubated at 37 °C in 0.5% CO_2_ (Panasonic, Japan). Embryos that died after microinjection under a stereo microscope (Leica, Wetzlar, Germany) were separated, and surviving embryos were transferred into M2 medium (Sigma, Darmstadt, Germany). Living embryos were transferred to the foster mother (healthy female with a vaginal plug after mating with vasectomized male mice), under anesthesia with a mouth pipette into the oviduct. Three weeks later, the F0 generation puppies, born 21 days after the intervention, were separated from their mothers according to sex, transferred to new cages, and simultaneously subjected to behavioral and molecular tests with other groups at two months of age (see in Appendix A for mouse groups).

### 2.5. Behavior Tests

Behavioral experiments were started when miRNA-microinjected and control group mice were 2 months of age. First of all, after the mice were two months old, they were mated to test whether psychological disorders or autistic behaviors occurred in the offspring (F1), as in our human family sample. In parallel, the F0 generation was subjected to behavioral experiments. After the behavioral experiments, the F0 generation was sacrificed. The F1 generation was evaluated only for behavioral changes. Each mouse underwent a single test daily between 10:00 and 16:00 h. Only males were tested sequentially on the same day in separate sub-sessions to allow room ventilation and cleanup. The testing tools were cleaned between trials with 70% ethanol and aerated before use. Experiments were videotaped and analyzed offline. The analysis of mouse behavioral tests, including sociability, social preference, object recognition, and tail suspension tests, was performed using the “EthoVision 9” software (Noldus, Wageningen, The Netherlands). Marble burying tests were analyzed manually by an observer blind to the group of mice.

#### 2.5.1. Novel Object Recognition Test

A mouse-size object and a second identical object were placed in a square box with an open top with lines dividing it at the base and a wall enclosing it. A mouse-sized object and a second identical object are placed. The mouse’s proximity to the objects and the number of visits were counted on the first day. On the second day, one of the objects was replaced with a new object. The discriminating index, the number of times the mouse approached both unfamiliar and recognized objects, and their proximity were all measured. The data obtained are defined as the discrimination index when the difference between the total time spent with the new object and the total time spent with the familiar object is divided by the total duration and multiplied by 100. The proportion of the mice’s overall interaction time with the new object is measured. The learning and memory abilities of mice are examined in this experiment. Because mice have an innate desire for novelty, it was anticipated that they would spend more time with the novel object to discover more about it [22].

#### 2.5.2. Social Interaction Test

The social test assesses cognition in mouse models of central nervous system (CNS) disorders by evaluating general sociability and interest in social novelty. Since they are sociable creatures, rodents normally prefer to spend more time with other rodents (sociality) and are more likely to approach a stranger than a friend (social novelty). The Crawley’s friendliness and social novelty test employed a rectangular three-compartment box, defining the experimental space for the study. Each area is 19 × 45 cm and is a system with partitioned walls and a central chamber made of clear plexiglass that allows open access to every area [23]. A mouse was initially placed into a middle compartment for 5 min while the other compartment was left empty. Under the parameters found to be connected to the mouse in the chamber, data were generated using the EthoVision system, and statistical analysis was performed along with comparisons to the control group.

#### 2.5.3. Marble Burying Test

The marble burying test is commonly used to evaluate rodent neophobia, which includes shyness around unfamiliar objects, anxiety, and obsessive-compulsive or repetitive behaviors [24,25]. We positioned the bedding that we usually use to care for our mice at the height of 5 cm in an empty cage. There are five rows of four marbles each, totaling twenty marbles. The experimental mouse was let out of one corner of the cage and given 30 min to roam around. The mouse was removed from the cage after 30 min, and the number of balls discovered under the bedding was tallied and recorded. The number of embedded balls was used for statistical analysis.

#### 2.5.4. Tail Suspension Test

The mouse suspended by the tail test is an experimental model of autism; on observing that after initial escape-directed movements, mice develop a sedentary stance when placed in an unavoidably stressful situation. The stressful situation during tail-hanging includes the hemodynamic stress of hanging, and autistic models seem to have reduced mobility and escape abilities [26]. The experimental setup was designed to simultaneously test three different mice. Thick cardboard-like sheets measuring 25 cm high were cut and placed between the mice so that they could not see each other. Since the mice are white in color, a black background was used. Twelve cm-long tapes were cut and hung in the experimental setup by sticking them to the tail ends of mice in such a way that their tails would not be damaged. By watching recordings with a video camera, the mobility and immobility time of mice were calculated for six minutes. The immobility time was used for statistical analysis.

After the behavioral experiments of the F0 generation were completed, two females and one male were mated to obtain the F1 generation. When obtaining the F1 generation, the F0 generation and the control group were sacrificed and blood, hippocampus, and sperm samples were taken. For the F1 generation, behavioral experiments were conducted when they were 2 months old. The F1 generation was only studied for behavioral changes and molecular analyses were not performed.

### 2.6. RNA Isolation from Tissue

After the behavioral experiments were completed, the mice were sacrificed and sperm and blood samples were collected. Then, the collected sperm and blood samples were placed in 500 µL of Purezol (Biorad, CA, USA, Cat No: 7326890). Subsequently, total RNA isolation was carried out according to the manufacturer’s instructions.

### 2.7. cDNA Preparation and Quantitative Real-Time Polymerase Chain Reaction (qRT-PCR)

Isolated RNA samples were reverse-transcribed into cDNA in 20 μL final reaction volumes using miScript II RT Kit (Qiagen, Hilden, Germany, Cat. No. 218193) as specified in the manufacturer’s protocol. Reverse transcription was performed using the SensoQuest GmbH Thermal Cycler (Göttingen, Germany). cDNA samples were kept at −80 °C until PCR analysis (see Appendix A for oligos sequences). qRT-PCR was performed by using miScript SYBR^®^ Green PCR Kit (Qiagen, Hilden, Germany, Cat No: 218073) with the high-throughput Light Cycler 480 II Real-Time PCR system (Roche, Mannheim, Germany). cDNA samples were diluted with nuclease-free water (1:5). The reaction was performed according to the manufacturer’s instructions. About 10 μL Syber Green Master Mix, 2 µL 10× Universal Primer, 2 µL primer assays, and 4 µL nuclease-free water were mixed and pipetted into a 96 well plate in a volume of 18 µL, and 2 μL of 1:5 diluted cDNA was pipetted into each well and mixed. The real-time PCR step was performed by using the Light Cycler 480 II Real-Time PCR system with the following protocol: thermal mix followed by the activation step at 95 °C for 15 min, then a denaturation step at 94 °C for 15 s followed by an annealing step at 55 °C for 30 s, and finally an extension step at 70 °C for 30 s. After the activation step, all steps were carried out for 40 cycles. Data normalization was performed using the 2^−ΔΔCT^ method with U6 internal control.

### 2.8. Data Analysis

After the results were obtained, experimental groups and control groups comparisons were made. The compliance of the data to normal distribution was evaluated using the histogram, q-q graphs, and Shapiro–Wilk test. We conducted detailed statistical analyses using a two-tailed, one-way analysis of variance (ANOVA) method, followed by the uncorrected Fisher’s least significant difference (LSD) test. This approach allowed us to thoroughly examine and compare the data, considering various factors and identifying any significant differences between groups in our study.Kruskal–Wallis, Student’s *t*-test, and Mann–Whitney U tests were also carried out depending on whether the data showed normal distribution or not. Data were analyzed using SPSS version 22 (IBM, Armonk, NY, USA) and Graph-Pad Prism 8.0 software(Prism, Boston, MA, USA). Results with *p* values < 0.05 were considered statistically significant. Data are expressed as the mean with SD.

### 2.9. Immunochemistry

Different mice were deeply anesthetized and transcardially perfused with 4% paraformaldehyde. A series of x-mm sections were incubated in a solution of mouse anti- (1:1000), rabbit anti- (1:500), goat anti- (1:500), and rabbit anti-GFAP (1:500) for 48 h at 4 °C. After several rinses in phosphate-buffered saline of x-conjugated, highly cross-adsorbed donkey anti-goat, (all at 1:500; Life Technologies, Carlsbad, CA, USA) in PBST. Sections were rinsed intensively and mounted onto slides, coverslipped using Mowiolmixed with Hoechst (1:10,000; Life Technologies), and stored at 4 °C.

The images were obtained with the DP71 digital camera under the B×53 light microscope (Olympos, Tokyo, Japan). JPEG images were imported into ImageJ/Fiji software (version 1.54h) (New York, NY, USA) to measure immunohistological staining for each protein. The threshold function was applied to separate the signal from the background and the average signal intensity was measured with the “measure” function.

## 3. Results

### 3.1. RNA Targeted Autism-like Mouse Model

Among the Turkish cohort of 37 families including children with autism (45 subjects altogether) [10], we were interested in a family with two affected children (see Appendix A). A father (normal) from two separate marriages has multiple children and one of each child from these separate unions develops a distinct psychological disorder. We chose this father because the six miRNAs, as reported in our previous study [10] were significantly downregulated by 50% of controls in this father’s blood, whereas in autistic patients, they represented 5% of controls. Additionally, the same miRNAs were also altered as well in the sperm sample of this father with two affected children (Appendix A). We examined during functional tests in mice whether the father’s sperm contained RNA responsible for the transmission of autism. If so, could RNA microinjection into fertilized mouse eggs also induce behavioral changes in adult mice born from these manipulations? Humans and mice, despite significant differences in DNA and gene expression, use a similar gene regulatory mechanism and network [27]. To investigate the potential control of non-Mendelian inheritance of ASDs in mice via RNA, we used the same approach previously used with mouse sperm RNA to evaluate human sperm RNA in a possible experimental model.

In addition, two-week-old young male mice were treated with valproic acid (VPA), and then sperm were collected from adult autistic mice (previously characterized in our laboratory) for functional analysis in mice.

Variations in transcripts play an essential role in brain development [3,28,29,30,31,32]. Therefore, when establishing an RNA-induced autism-like mouse model, we exposed fertilized mouse eggs to exogenous total RNA from sperm (see Appendix A for the experimental plan).

Previous analyses in human sera of six miRNAs (-5p and 3p strands of the same miRNA) and in one sample from the father’s sperm showed that all were altered, with most being downregulated. However, in one of the father’s patients available, two sperm RNAs (miR-499-3p and miR-361-5p) were upregulated to exaggerated levels. Mice born after microinjection of paternal sperm RNA again showed significant downregulation of four miRNAs (361-3p, 150-5p, 126-5p, and 499-3p), of the five present in mice. Paternal sperm RNA (father of patients) negatively regulates at least four miRNAs in the blood of adult mice (Figure 1).

Adult blood samples were collected when the mice were sacrificed as adults (three months). We analyzed to examine the levels of specific miRNAs (miR-19a-3p, miR-19b-5p, miR-19b-3p, miR-126-5p, miR-126-3p, miR-499a-5p, miR-499a-3p, miR-361-5p, miR-361-3p, and miR-150-5p) in the blood of mice. The miRNAs were investigated individually, as illustrated in Figure 1A–J. 

The two miRNAs upregulated in paternal sperm miR-361-5p (forward of 3p) and miR-499-3p (reverse of 5p) are now downregulated in mice, as it was in children (patients) [10]. Whereas, as mentioned above, these two miRNAs (miR-361-5p and miR-499-3p) are abnormally upregulated in the father’s sperm [10]. This may suggest that miRNAs at exaggerated levels of upregulation in sperm trigger downregulation in the next generation (see Figure 2). 

Sperm samples were collected during adult sacrifice. Analysis of miRNAs expression levels in mouse sperm, A to J (miR-19a-3p, miR-19b-5p, miR-19b-3p, miR-126-5p, miR-126-3p, miR-499a-5p, miR-499a-3p, miR-361-5p, miR-361-3p, and miR-150-5p, respectively) was performed. Mice received RNA from the patient’s father’s sperm and/or RNA from healthy sperm via microinjection.

### 3.2. Behavioral Variations

Genes linked to ASDs are also found in mice and have been analyzed by targeted mutagenesis. Crucially, so far, all genetic mutations linked to ASDs also perturb brain development when tested in mice (review by Crawley [33]). Although all the deficits defining autism are difficult to assess in mice, significant efforts are being made to develop behavioral tests to phenotype mice. Based on the behavioral neuroscience literature, we examined mouse colonies on existing behavioral tasks to refine key symptoms. The summary schedule for the behavioral task analysis is listed in Appendix A.

Behavior tasks were carried out on male mice from the age of 2 months through recognition of a novel versus familial object, social interactions, suspension by the tail, and marble burying assays (see Figure 3). While it was observed that the interest in the new object was higher in all groups in the novel object recognition test, no significant difference was detected between the groups in terms of discrimination index, total movement, and speed. In the social interaction test, all groups, except the control group, including the F0 and F1 generations, showed more interest in the empty cage and exhibited behaviors that did not support social life and lacked empathy. In the marble test, there is an increased marble burying activity towards the next generation compared to the control, which is an indicator of hippocampal damage. This situation is clearly observed in the mouse group born after the microinjection of RNA from the sperm of the patient’s father. In the tail suspension test, where anxiety findings were measured, it was determined that although the most inactive group was the F1 born from the human sperm RNA control group, all groups exhibited similar phenotype behavior to the control.

### 3.3. Histological Examinations

The prefrontal cortex, corpus callosum, striatum, hippocampus, amygdala, and cerebellum are widely recognized as regions associated with autism. In all groups, brains were studied in detail with histochemical stains such as hematoxylin–eosin and luksol fast blue (Figure 4, Figure 5 and Figure 6). The structure of the corpus callosum in the brains of all groups of animals was examined in the striatum and hippocampus regions and found to have a normal structure, as in the brains of animals in the control group. No agenesis of the corpus callosum was reported in any of the experimental groups observed in the brains of autistic individuals. The lateral ventricles and third ventricle were examined in the brains of all groups of animals and the dimensions of the ventricles were determined to be normal as in the brains of the control group animals. In other words, ventricular dilation, which is believed to be present in the autistic human brain, was not observed in animals from all experimental groups. 

Using histochemical stains such as hematoxylin–eosin and Nissl, the brains of the experimental groups were extensively examined in regions thought to be associated with the pathophysiology of autism, including the prefrontal cortex, corpus callosum, the striatum, the hippocampus, the amygdala, and the cerebellum. The structure of the corpus callosum, as well as that of brain regions such as the striatum and hippocampus, was studied in the experimental groups and found to be similar to that of the control group. In the experimental groups, conditions such as agenesis of the corpus callosum, ventricular dilatation, and ectopic white matter bundles reported in the autistic brain were not observed. The layers of CA1, CA2, and CA3, and the dentate gyrus in the hippocampus were found to have a similar structure to that of the control group. At the cellular level, no neurodegeneration was observed in regions of the prefrontal cortex, striatum, hippocampus, and cerebellum.

At both anatomical and histological levels, no specific changes were observed in the cerebellum and brainstem when comparing animals in all experimental groups to those in the control group. Furthermore, there was an absence of white matter substances between the hemispheres in the region of the striatum and the hippocampus along the midline in all groups.

It was observed that the outer capsule, inner capsule, and anterior commissure in the brains of all groups of animals did not present a different status from that of the brains of animals in the control group. It was determined that the sizes of the striatum and septum, as well as the cortical thicknesses of animals in all groups did not differ from those of control mice.

No differences were observed in the brains of all groups of mice compared to the brains of control animals in layers CA1, CA2, and CA3, and from the dentate gyrus to the hippocampus. Similarly, no discernible distinctions were noted in the anatomical and cellular structure of amygdalin in the brains of all groups of mice compared to the brains of animals in the control group. Furthermore, there was no reduction in the crescent size of the hippocampal in the brains of all groups of mice.

At the cellular level, neurodegeneration was not observed in the prefrontal cortex, striatum, hippocampus, and cerebellum regions of animal brains in all groups, but gliosis was observed in the VPA, human sperm RNA, and reverse miRNA groups (Figure 4, Figure 5 and Figure 6 and Appendix A). No ectopic white matter clusters, as reported in autistic human brains in cingulate cortexes and other cortexes, were found in the brains of all groups of mice. Histological examination of the cerebellum in all groups of mice did not reveal areas of Purkinje cell loss which has been reported in the autistic human cerebellum.

### 3.4. Immunohistochemistry

The prefrontal cortex, corpus callosum, striatum, hippocampus, amygdala, and cerebellum, suspected to be associated with the pathophysiology of autism in the brains of animals of all groups, were studied via histochemical immunostaining. Differences were observed in the VPA, miR-reverse, and total-RNA groups. Astrocytes displayed abnormal features characterized by astrogliosis (and microgliosis, which is normally invaded by microglia). However, it is important to note that no instances of astrocyte misorientation or microglial extension as reported in the human brain with autism, were observed. Furthermore, there was no evidence of hypertrophy or hyperplasia of GFAP and Iba-1 cells in immunopositive cells (Figure 7, Figure 8, Figure 9, Figure 10 and Figure 11).

Fluorescent staining in the hippocampus, cerebellum, and striatum of mice born after microinjection of RNA from sperm of patient’s father.

We conducted GFAP immunostaining on samples from the prefrontal cortex, striatum, hippocampus, and cerebellum. In Figure 9A, GFAP immunostaining images at 10×, 20×, 40×, and 100× scales are shown for various experimental groups. The corresponding data in the histogram (Figure 9B) are presented with mean bars and SD. Additionally, the control RNA from human sperm and the reverse miR499a-5p group are different from the control group and the total RNA from the sperm of the patient’s father.

Microglia cells and astrocytes are known to play an important role in regulating neuronal metabolism, providing nutritional support to neurons, shaping, facilitating synapse formation, and contributing to neurotransmission. Furthermore, in the mouse group born from mice microinjected with RNA from human sperm (from both the father of autistic children and the normal group), the number of oligodendrocyte products was increased and significantly branched compared to the control and VPA groups. So, at this point, these results reveal an additional side effect of human sperm RNAs rather than just autism. Unlike the miRs previously examined, miR-499b-3p also induced the same changes as observed in sperm total RNAs and VPA groups in astrocytes and microglia, although miR-19a and miR-499b-5p did not produce the same effects.

## 4. Discussion

The present study revealed the broad regulatory function of sperm RNA in developing mouse embryos even from different species including that of sperm RNA derived from the father of an autistic child. With the microinjection of RNA from the sperm of the father of an autistic child into fertilized mouse eggs, we revealed here a program capable of interfering with the development of the mouse embryos, according to which it was possible to establish an “ASD-like” mouse model. Analysis of groups of mice born after manipulation revealed downregulation of autism-related miRNA in blood and hippocampus with affected glial cells accompanied by changes in behavioral phenotypes. The RNA transferring the development of autism is present in the father’s sperm as a non-Mendelian hereditary information, since an altered phenotype could be established in mice. These results suggest that instructions for RNA variations, such as downregulation of a group of distinct miRNAs are transferred and may contribute to the development of ASDs. miRNAs are transcribed by polymerase II [14,34,35], but in developing embryos, the regulation of transcription of miRNA or a distinct group of miRNA altogether is not yet known.

Furthermore, there was a correlation between the effects induced by valproic acid (VPA) and those induced by the patient’s father’s sperm RNA, suggesting shared pathways affected by both the mouse VPA model and sperm RNA (human)-induced phenotype. Although glial cells are thought to play a major role in brain development and pathogenesis in general, concrete evidence for their initiating role in autism was lacking, as such analyses were typically conducted post mortem. Here, with the sperm RNA transfer assay, we induced gliosis and astrogliosis for the first time, thus confirming the working hypothesis of their impaired function in ASD. It is important to pay attention to early disturbances induced from the single-cell stage, as reported here that alterations induced at this early stage persist and influence the possible development of ASD symptoms. In particular, our study in mice suggests that glial cells are the first affected. A recent review clearly highlights the different functions of glial cells in autism [20].

Although comparing the age of mice/or other animals to humans is not always appropriate, a young adult mouse could provide a first glimpse into the consequences of altered gene expression during embryonic development. The two-month period was necessary to obtain comprehensive results from behavioral tests and histological analysis. Conducting behavioral testing before puberty, especially with early separation from the mother’s littermate could potentially affect their behavior.

In this study, we examined all sections of the brain with careful analysis of regions reported in previous studies. However, the most prominent observed alteration was related to glial cells and this effect was consistent between the VPA and exposure to RNA from autistic progenitor sperm or through microinjection of miR-499-3p reverse miRNA, while miR-499-5p did not induce gliosis or astrogliosis.

The characterization of trans-species (human sperm) RNA-induced changes in mice can reveal networks of RNA-mediated signals. Our study not only confirmed the previously proposed non-Mendelian RNA-mediated ASD-like phenotype in the 2-month-old mouse model, but also confirmed RNA alterations in ASD development. Autism-linked miRNAs were downregulated in the blood, hippocampus, and sperm of two-month-old mouse models, along with corresponding changes in behavior. This suggests a potential association between altered miRNA expression markers in blood and changes in glial cell histology and phenotype. Indeed, as recently reported, glial cell signaling was one of the most affected pathways in NDDs [20]. Alteration of glial cells accompanied by molecular and phenotypic changes is consistent with the importance of early developmental stages and so the critical period theory of ASD. The identification of glial cells in the mouse model is particularly important, as it supports that ASDs are highly influenced by a mechanism involving the regulation of synaptic formation and function by glial cells (par example removal of debris by phagocytosis, etc.). Other changes were also noted with paternal RNA, but were also observed in control exposed to human sperm RNA, and they require further investigation.

Defining molecular functions at an early stage and finding an appropriate timing for intervention, could be an important way to prevent the development of ASDs. The present study revealed that VPA mouse models exhibit the same molecular, glial, and behavioral phenotypes that are also induced by sperm RNAs from the father of a child with ASD. In particular, the comparable concordant RNA activity between mouse (VPA) and human ASD progenitor suggests that the mouse model is a suitable biological system for testing pharmacological components in understanding ASD.

Our previous studies on autistic behavior coupled now with the findings of morphological alterations in glial cells, reaffirm that mice with ASD have continued to serve as a practical model for translational study. Furthermore, this study opens up several avenues for exploration. Firstly, routine testing of the father’s sperm from individuals with different etiologies (see below) and symptoms could be considered. Secondly, we highlight and confirm with the literature the roles of non-neuronal components such as astrocytes and microglia, whose accumulating evidence suggests involvement in the pathogenesis of ASD. Thirdly, the detection of ASD-related markers at two months of age in the mice allows for additional screening methods and for testing treatment. That is particularly valuable as medications intended for newborns and children generally carry unwanted side effects and their effectiveness may not be fully established.

MicroRNA (miRNA) alterations are observed in different diseases. In a comprehensive investigation involving 45 autism spectrum disorder (ASD) patients and 21 controls, our results reveal the potential diagnostic significance of downregulation of miR-126-3p, identified as a prognostic predictive biomarker through machine learning methods [36]. In a parallel study using the Marmarau trauma model in mice subjected to head trauma, we explored the expression levels associated with miR-126-3p. Our investigation unveiled the impact of miR-126-3p on the hypothalamus–pituitary–adrenal (HPA) axis, leading to increased release of Adrenocorticotropic Hormone (ACTH). These results highlight the protective effect of miR-126-3p on the HPA axis, also providing a potential role as a neuroprotective agent in the context of traumatic conditions [37].

The precise mechanism underlying age-related oxidative stress and its role in the aging process remains unclear. It is hypothesized that increased levels of reactive oxygen and nitrogen species (RONS) likely trigger cellular senescence, a natural mechanism halting cell proliferation in response to replication-induced damage. Although the impact of advanced maternal age is well-documented, recent research is increasingly focused on understanding the implications of advanced paternal age (APA) on reproduction. With age, the antioxidant defense system gradually declines in effectiveness, leading to the accumulation of high levels of reactive oxygen species (ROS). This accumulation can harm the functional and structural integrity of germ cells including sperm. Male aging shows a negative correlation with various sperm parameters, reproductive hormone levels, testicular function, chromosomal structure, and sperm DNA integrity. These factors collectively contribute to infertility and can have detrimental effects on offspring [38,39].

Also, the researchers investigated the impact of oxidative stress on offspring by examining changes resulting from exposure of sperm to hydrogen peroxide (H_2_O_2_). Treating sperm with H_2_O_2_ led to adverse effects on embryo development and diminished fetal growth. These effects were particularly noticeable in female offspring, highlighting alterations in body composition and disruptions in glucose regulation [40].

Oxidative modification of RNA has also significant consequences on the translational process, leading to disruptions in protein synthesis that can result in cellular deterioration or even cell death. Hydroxyl radicals (OH), formed in close proximity to RNA, readily engage in the modification of RNA molecules owing to their highly reactive nature and limited diffusion away from their sites of origin [41].

The occurrence of RNA oxidation is associated with various neurological diseases, such as Alzheimer’s disease, Parkinson’s disease, amyotrophic lateral sclerosis (ALS), spinal cord injury, and epilepsy. Numerous studies suggest that RNA oxidation represents an early event in the pathological progression of neurodegeneration. For instance, in an ALS mouse model, heightened RNA oxidation is observed, primarily in the motor neurons and oligodendrocytes of the spinal cord, starting as early as one month of age. This oxidative modification progressively increases with age and subsequently decreases as motor neurons begin to degenerate. These findings collectively emphasize that RNA oxidation is an early event preceding neuron degeneration and is not merely a consequence of cell death [41].

DNA methylation plays a crucial role in maintaining tissue-specific gene expression and genomic imprinting. DNA methylation variations upon environmental changes have been implicated in health issues and increased susceptibility to complex diseases. Understanding the intricate interplay between DNA methylation and oxidative stress is essential for unraveling the underlying molecular mechanisms that contribute to the development and progression of these complex health conditions [42].

The MTHFR enzyme plays a crucial role in the one-carbon metabolism (OCM) process, which encompasses the metabolisms of folate and homocysteine (Hcy). It is responsible for converting 5,10-methylenetetrahydrofolate (5,10-methylene THF) into 5-methyltetrahydrofolate (5-methyl THF). This enzyme is integral to the conversion processes of folate and homocysteine, which are closely tied to DNA methylation. Researchers have identified a significant association between the *MTHFR C677T* polymorphism and both schizophrenia and major depression within the general population [43]. Furthermore, in the recessive model, this polymorphism has been linked to an elevated risk of bipolar disorder. In a separate study, although not statistically significant, researchers observed a higher frequency of the *MTHFR 677T-allele* in autistic children compared to nonautistic children [44]. Several studies have demonstrated a connection between elevated homocysteine levels, reduced levels of vitamins B12 and folate, and autism spectrum disorder (ASD). These disruptions may be attributed to genetic predispositions such as *MTHFR* gene polymorphisms, potentially playing a role in the development of ASDs.

Furthermore, in males (mice) exposed to oxidative stress, transcription is altered in organs, notably spermatozoa [45]. Then, the embryos resulting from fertilization by these spermatozoa present modifications in adiposity and glucose regulation, particularly in the female offspring [38,40]. It is now important to test the sperm of patients’ fathers for a number of criteria related to oxidative stress. For example, MTHFR-SNPs/homocysteine impacts NDDs [46,47]. One possibility is that fathers of multiple children with ASD carry *MTHFR/SNPs* [48] that induce transcriptional alteration. Oxidative stress impairs the regulation of DNA methylation and is also involved in metabolic syndrome [46].

RNAs are the regulators of pre-implantation development because they prevent rapid/immediate translation of stored RNAs during oocyte growth [49,50]. In contrast, at the same time, methylation maintenance activity is only strong until the maternal–zygotic transition by *DNMT1* [51,52]. However, oxidative stress increases with age but could also occur during childbirth, suspected of being one of the important risk factors for autism [53] (see review [41,54]).

Furthermore, our results recall multiple cases of small RNA variations interfering with development, responsible for the alteration of alleles in trans with phenotypic changes in the following generation, notably in organisms such as maize [55]. Similarity of alterations of transcription and negative regulation in different organisms induced by RNA highlights, on the one hand, the recognition of the functional role of RNA in non-Mendelian heredity, and on the other hand, a mechanism importantly required during evolution.

Nonetheless, this study indicated the crucial importance of RNA-induced phenotypes as heritable factors in ASDs. These results could contribute to future studies on how to track RNA-mediated diseases and plan new therapeutic approaches for ASD early in life.

## Figures and Tables

**Figure 1 biomolecules-14-00201-f001:**
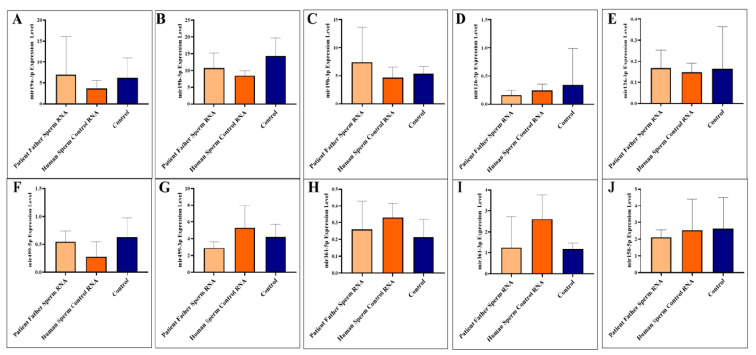
Analysis of miRNA expression in mouse blood (**A**–**J**). The mice were subjected to microinjection with RNA derived from either the sperm of the patient’s father or RNA from healthy sperm.

**Figure 2 biomolecules-14-00201-f002:**
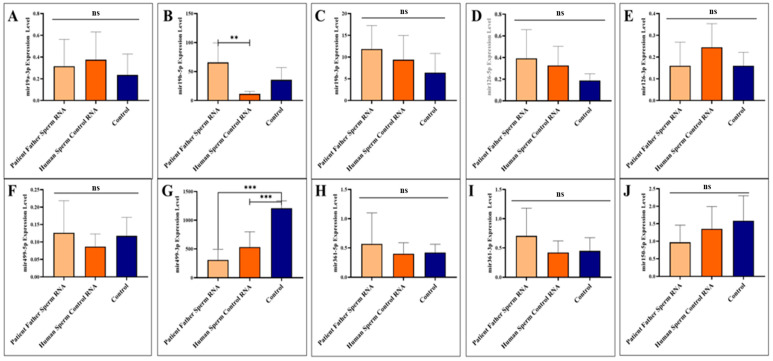
Analysis of miRNA expression in mouse sperm (**A**–**J**). Significant differences were denoted by ns = *p* > 0.05, ** *p* ≤ 0.01, *** *p* ≤ 0.001.

**Figure 3 biomolecules-14-00201-f003:**
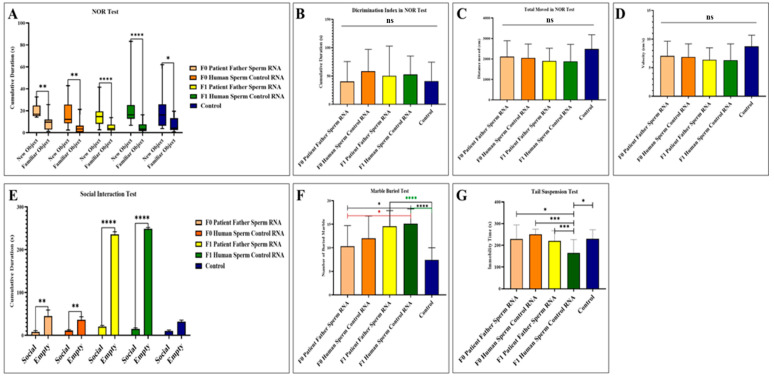
Results of behavioral experiments performed in adult mice to which RNA from the patient’s father’s sperm RNA or RNA from the healthy father’s sperm were transferred via microinjection. The control is a group of uninjected mice. (**A**) Measurement of the mice’s interest in the novel object recognition test. (**B**) Novel object recognition test (NOR) showing the difference in discrimination index between groups. (**C**) Demonstration of the distance traveled by mice during the recognition test of a novel object. (**D**) Demonstration of the speed of mice during the recognition test of a novel object. (**E**) Demonstration of social interest and interest of the mouse in the empty cage during the social interaction test. (**F**) Display of the number of marbles buried by the mice during the marble buried test. (**G**) Illustration of the time spent inactive by the mice during the suspension test by tail. F0 father (human) sperm RNA group, *n* = 9; F0 human sperm control RNA group, *n* = 10; F1 father (human) sperm RNA group, *n* = 23; F1 human sperm control RNA group, *n* = 16; and non-injected control group, *n* = 15. Significant differences were denoted by ns = *p* > 0.05, * *p* ≤ 0.05, ** *p* ≤ 0.01, *** *p* ≤ 0.001 and **** *p* ≤ 0.0001.

**Figure 4 biomolecules-14-00201-f004:**
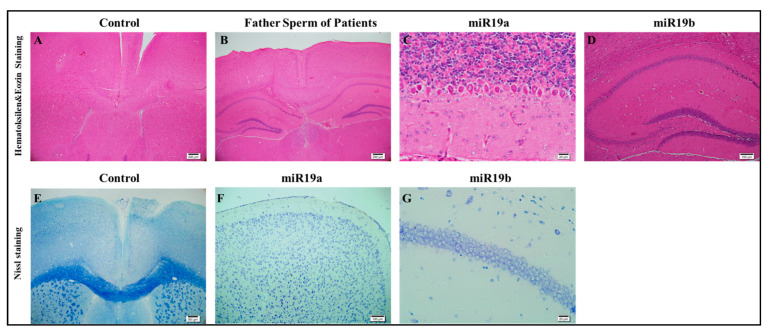
Histopathology (**A**–**G**).

**Figure 5 biomolecules-14-00201-f005:**
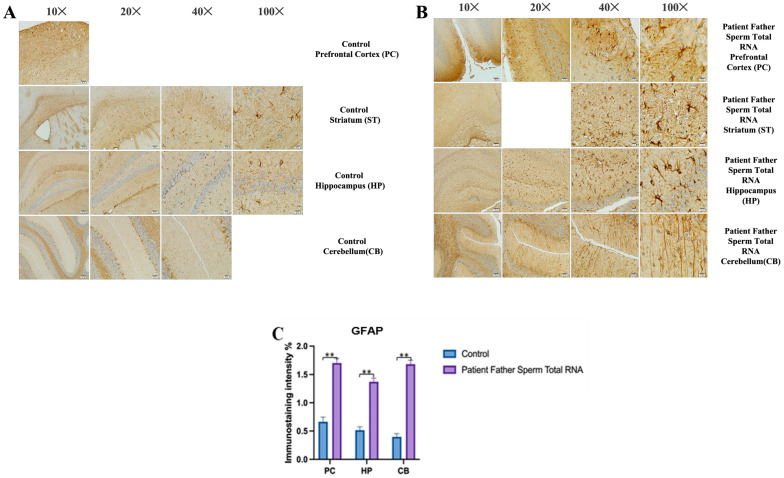
GFAP immunostaining images of prefrontal cortex, striatum, hippocampus, and cerebellum samples belonging to the experimental groups. (**A**) GFAP Immunostaining images in 10×, 20×, 40×, and 100× scale in the control group (non-injected). (**B**) GFAP Immunostaining images in 10×, 20×, 40×, and 100× scale in patient father sperm total RNA group. (**C**) Data shown in the histogram are arranged with mean ± SD bars. Independent *t*-test was applied. Significant differences were denoted by ** *p* ≤ 0.01.

**Figure 6 biomolecules-14-00201-f006:**
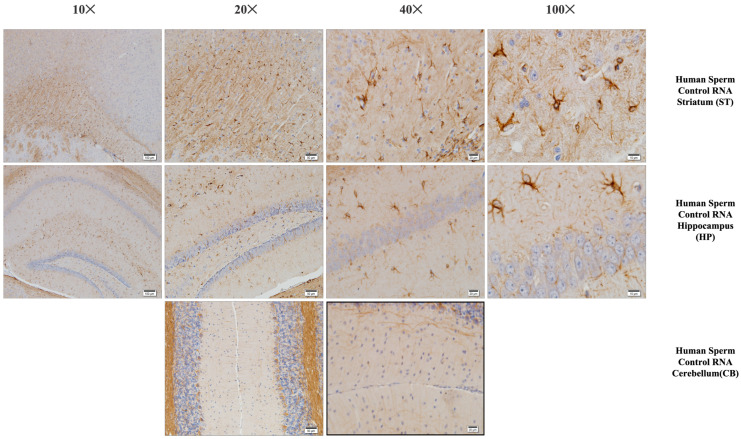
Iba-1 immunostaining of the prefrontal cortex, striatum, hippocampus, and cerebellum (mouse brain). Histological examination of the mouse brain is shown, a group of mice born after microinjection (into fertilized mouse eggs) of RNA from human sperm control.

**Figure 7 biomolecules-14-00201-f007:**
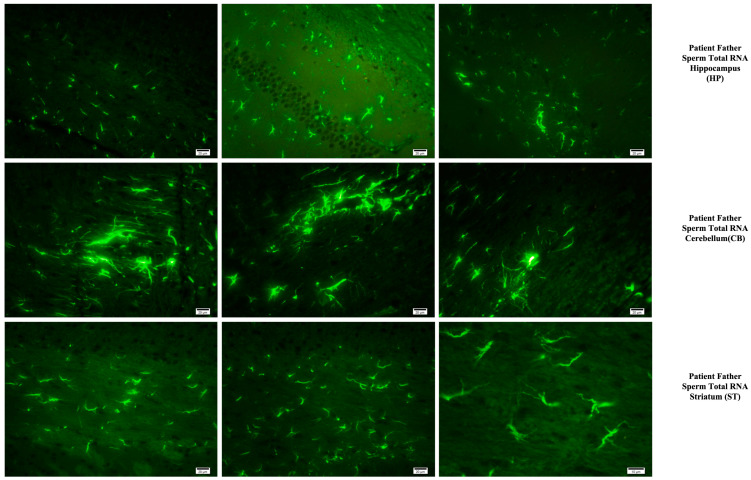
Immunostaining with mouse GFAP antibody in mice.

**Figure 8 biomolecules-14-00201-f008:**
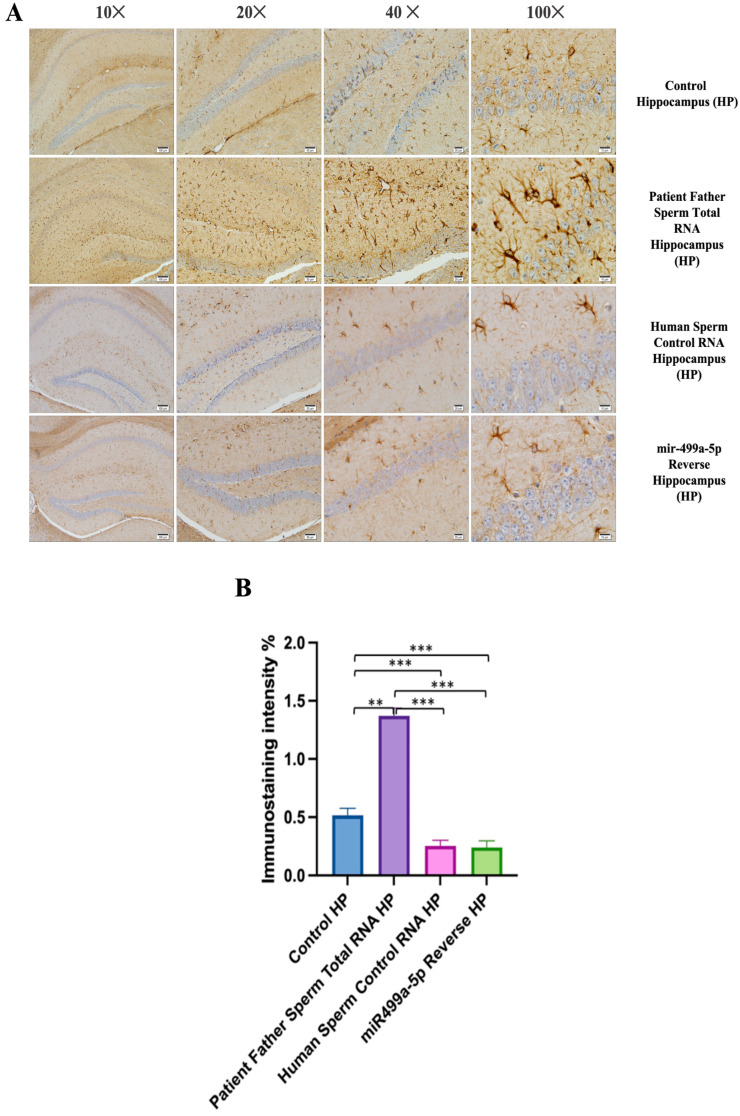
Immunostaining with mouse GFAP antibody in different groups of mice (**A**,**B**). (**A**) displays immunostaining images at 10×, 20×, 40×, and 100× scales for various experimental groups. (**B**) displays the data in a histogram format, with mean bars and standard deviation (SD). The statistical analysis involved one-way analysis of variance and Tukey’s multiple comparison tests. Significant differences were denoted by ** *p* ≤ 0.01, *** *p* ≤ 0.001.

**Figure 9 biomolecules-14-00201-f009:**
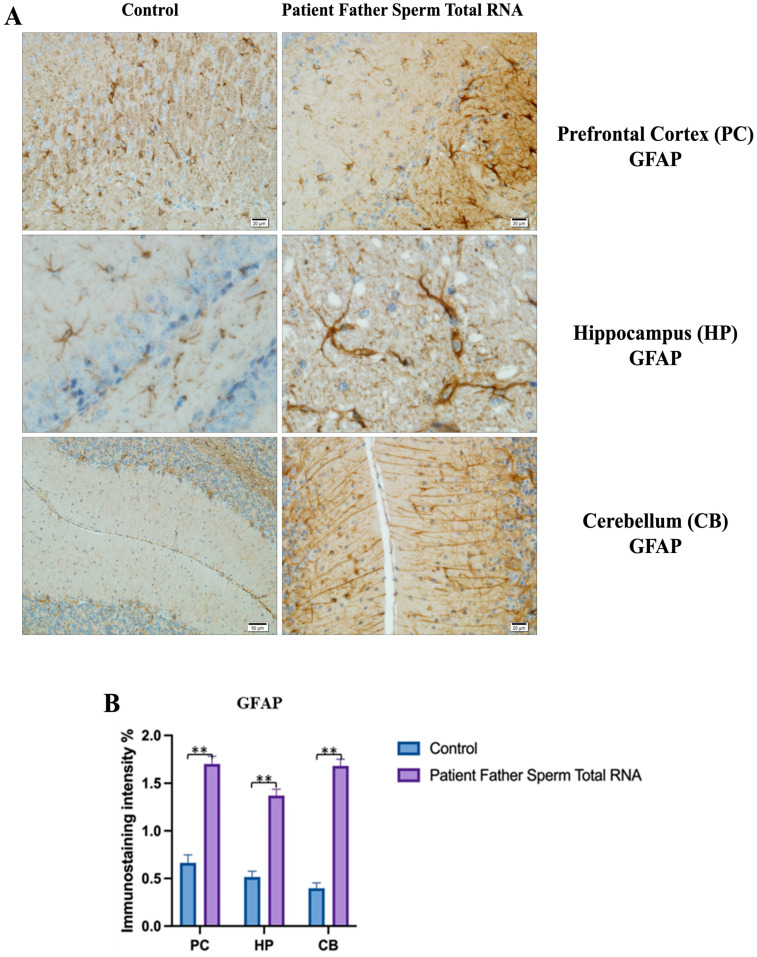
Visualization of GFAP immunostaining images of prefrontal cortex, hippocampus, and cerebellum. (**A**) GFAP immunostaining images of samples belonging to the control or those microinjected with sperm RNA from the father of patients in the experimental groups. (**B**) Data shown in the histogram are arranged with mean ± SD bars. An Independent *t*-test was applied. Significant differences were denoted by ** *p* ≤ 0.01.

**Figure 10 biomolecules-14-00201-f010:**
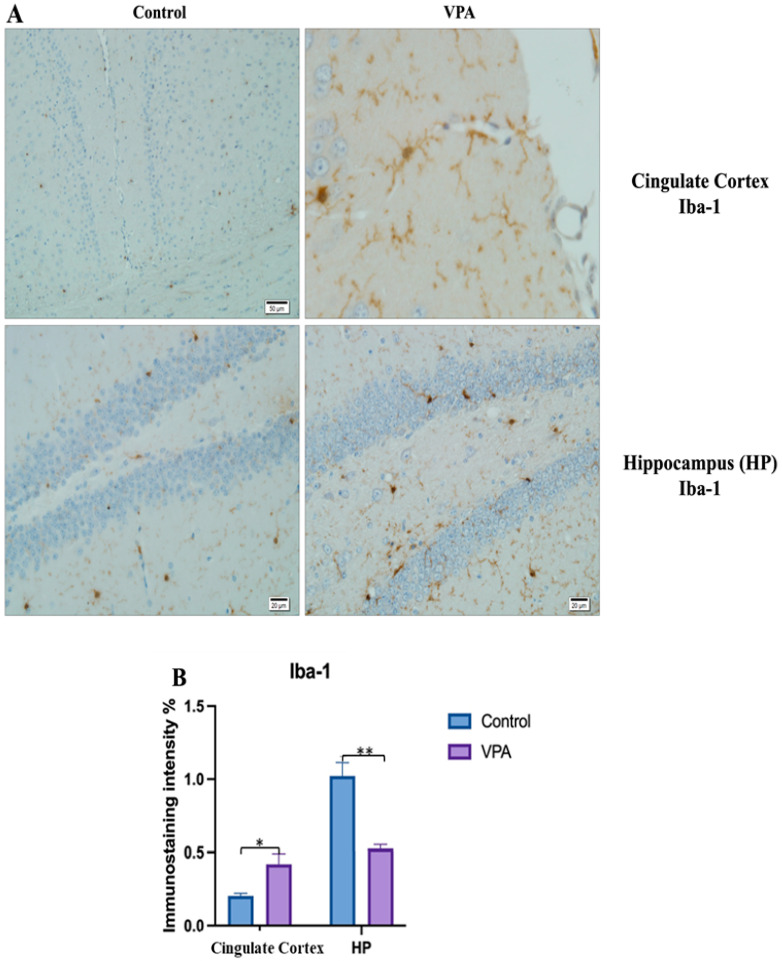
Visualization of Iba-1 immunostaining images of cingulate cortex and hippocampus samples. (**A**) Iba-1 immunostaining images of the control and VPA groups. (**B**) Data shown in the histogram are arranged with mean ± SD bars. Independent *t*-test was applied. Significant differences were denoted by * *p* ≤ 0.05, ** *p* ≤ 0.01.

**Figure 11 biomolecules-14-00201-f011:**
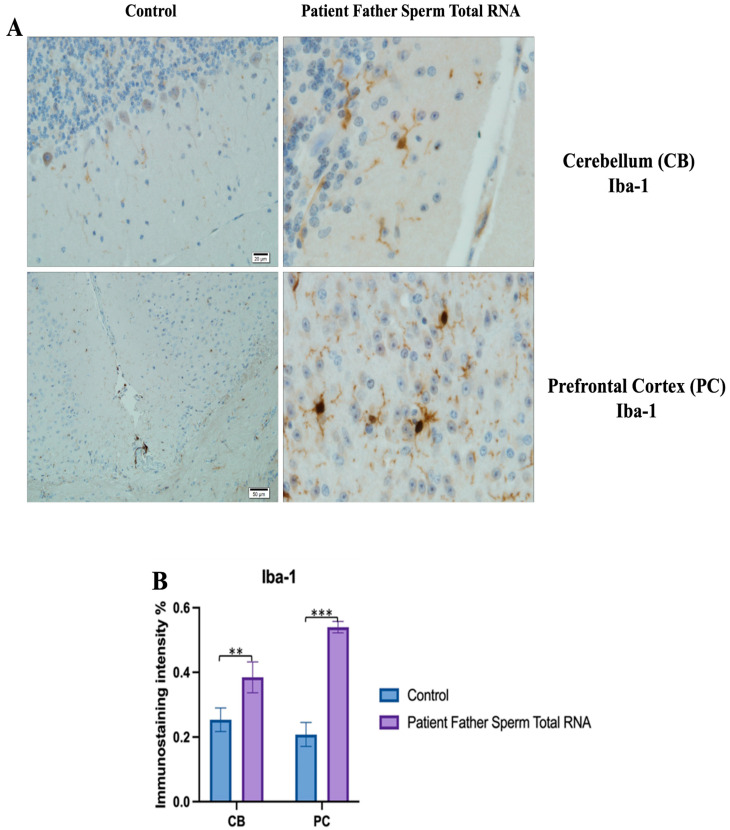
Visualization of Iba-1 immunostaining images of cerebellum and prefrontal cortex samples. (**A**) Iba-1 Immunostaining images belonging to the control non-injected and microinjected with RNA from sperm of the father of the patient experimental group. (**B**) Data shown in the histogram are arranged with mean ± SD bars. Independent *t*-test was applied. Significant differences were denoted by ** *p* ≤ 0.01, *** *p* ≤ 0.001.

## Data Availability

If the responsible authors are reached, all authors agree that all data can be shared with confidence, in line with the request of the researchers.

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
