# Peer review of "Trans Species RNA Activity: Sperm RNA of the Father of an Autistic Child Programs Glial Cells and Behavioral Disorders in Mice"

_biomolecules, 2024, doi:10.3390/biom14020201_

Round 1
Reviewer 1 Report
Comments and Suggestions for Authors
I recommend expanding the cited references list.
Minor writeup/revisions are needed to make results clearer.
Paper needs clearer definition of results and potential application to actual cases.
Author Response
Author:
We appreciate and thank the reviewers for their time, efforts to provide comments and criticisms. We have tried to clarify sentences and be as close as possible to the experiments.
-The English edition is improved, one of our American colleague make correction. If the manuscript is accepted, we will request help and English editing from the publisher.
Reviewer 1
I recommend expanding the cited references list.
Minor writeup/revisions are needed to make results clearer.
Paper needs clearer definition of results and potential application to actual cases.
Authors:
Totally agree, apologize for any mistakes and try to make it easier for readers.
Some errors were unintentionally introduced into the text when transferring the word text to the formatted page.
Reviewer 2 Report
Comments and Suggestions for Authors
The motivation for this study is based on prior investigation of miRNA in families with one or more autistic children. In this study the investigators found 6 miRNA's with ow or very low expression in serum from 45 children with autism; some reduction of the levels of the same miRNA's in parents or relatives, but "never in any unrelated control" [however, the control population of 16 parents and 21 children is small by current standards.]
Interestingly, the investigators found a similar profile in valproic acid treated mice and in a murine genetic model of autism.
There is some evidence of Mendelian expression of the levels of these miRNAs; 40-50% reduction in parents and some reason to believe that the same miRNA's low in sperm from transmitting parents.
The experiment described here essentially involves the possibility of gene transfer using human sperm as a vector to manipulate biology in fertilized murine ova. The specific question being examined is whether human sperm with low expression of the 6 target miRNA's will produce mice with "autistic-like" features. Briefly, total RNA was extracted from sperm of both carrriers and non-carriers of low expression of the target mRNAs and this was microinjected, at a level of 2 ng/µl (volume?) into fertilized mouse eggs. Progeny were identified as SH (control sperm) or ASH (progeny of ova treated with sperm from transmitting carrier).
The paper is poorly written (some examples follow) and more important, very weak statistically.
confusing or syntactically irregular examples: last sentence of ¶4 of introduction - lines 71-73
Last 2 sentences of next paragraph: how could behavioural traits cause a change in the number or structure of oligodendrocytes? isn't it more likely to be the other way around?
line 94: one of each bed?
sentence 110-112 seems to suggest that the observed behavioural effects result from the sperm treatment in general, rather than the treatment with sperm from the transmitting carrier.
RNA microinjection: what is meant by a "mouth pipette"?
many more throughout the manuscript
After the birth of offspring, behavioural tests were done and RNA from both blood and sperm was evaluated.
Results; the family pedigree is not a result, but rather the motivating factor.
"Progeny of the carrier treated animals showed significant down-regulation of 4 miRNAs" in blood: Figure 3 and sperm: Figure 4- no statistical analysis presented; the figure legend does not facilitate identification of which miRNA is being measured in which panel; visual inspection does not correlate with the alleged result and there is a confusing discussion about 2 upregulated miRNA's from parent not consistent with initial paper reporting on 6 downregulated miRNA's.
Are the control bars in these panels from mice not microinjected?
Several behavioural tests were performed in 5 groups of animals, labelled Control, F0 SH, F0 ASH, F1 SH, F1 ASH (nomenclature as in ¶2 of this review). Although F0 and F1 offspring appear in the cartoon (Figure 2) summarizing the experiment, this is not discussed in the methods. Again, there are bars in the panels of Figure 5 indicating significance, but no statistical analysis is presented, and the sample sizes of each of the animal groups for each of the behavioural tasks is not noted.
I have no concern with the photomicrographs presented as examples of histochemical analysis, but the same concerns about sample size and statistical analysis pertain here.
Throughout the manuscript there are awkward sentences and phrases and both the text and the reference list require careful reading and correction of both typographical and other errors: example, reference 15 comprises 2 references, and reference 16 refers to 2 different citations.
Comments on the Quality of English LanguageThe investigators obviously write many papers in English, but this one was either rapidly constructed or they normally have a reviewer who picks up the many lapses and typographical errors. This is less an issue of English per se, and more an issue of poor reviewing of the manuscript for readability, typos, and errors of statement.
Author Response
Author:
We appreciate and thank the reviewers for their time, efforts to provide comments and criticisms. We have tried to clarify sentences and be as close as possible to the experiments.
-The English edition is improved, one of our American colleague make correction. If the manuscript is accepted, we will request help and English editing from the publisher. Some errors were unintentionally introduced into the text when transferring the word text to the formatted page.
Reviewer 2
The motivation for this study is based on prior investigation of miRNA in families with one or more autistic children. In this study the investigators found 6 miRNA's with ow or very low expression in serum from 45 children with autism; some reduction of the levels of the same miRNA's in parents or relatives, but "never in any unrelated control" [however, the control population of 16 parents and 21 children is small by current standards.]
Interestingly, the investigators found a similar profile in valproic acid treated mice and in a murine genetic model of autism.
There is some evidence of Mendelian expression of the levels of these miRNAs; 40-50% reduction in parents and some reason to believe that the same miRNA's low in sperm from transmitting parents.
The experiment described here essentially involves the possibility of gene transfer using human sperm as a vector to manipulate biology in fertilized murine ova. The specific question being examined is whether human sperm with low expression of the 6 target miRNA's will produce mice with "autistic-like" features. Briefly, total RNA was extracted from sperm of both carrriers and non-carriers of low expression of the target mRNAs and this was microinjected, at a level of 2 ng/µl (volume?) into fertilized mouse eggs. Progeny were identified as SH (control sperm) or ASH (progeny of ova treated with sperm from transmitting carrier).
Authors
We specified the concentration (2ng/ µl) and the volume of the microinjection (1-2 picoliter).
Reviewer 2
The paper is poorly written (some examples follow) and more important, very weak statistically.
confusing or syntactically irregular examples: last sentence of ¶4 of introduction - lines 71-73
Last 2 sentences of next paragraph: how could behavioural traits cause a change in the number or structure of oligodendrocytes? isn't it more likely to be the other way around?
Authors
That is right we changed order.
Reviewer 2
line 94: one of each bed?
Authors
Woman or mother, modify sentences.
Reviewer 2
sentence 110-112 seems to suggest that the observed behavioural effects result from the sperm treatment in general, rather than the treatment with sperm from the transmitting carrier.
RNA microinjection: what is meant by a "mouth pipette" ?
Authors
small glass transfer pipette, routinely is named mouth pipette in laboratory and generally in the protocol, because one need to control handling embryos (80 micron).
Reviewer 2
many more throughout the manuscript
After the birth of offspring, behavioural tests were done and RNA from both blood and sperm was evaluated.
Authors
We apologize for any errors and unclear sentences, we have tried to improve.
Reviewer 2
Results; the family pedigree is not a result, but rather the motivating factor.
Authors
That's right, thanks for the point raised, Figures 1 and 2 have been transferred to Materials and Methods.
Reviewer 2
"Progeny of the carrier treated animals showed significant down-regulation of 4 miRNAs" in blood: Figure 3 and sperm: Figure 4- no statistical analysis presented; the figure legend does not facilitate identification of which miRNA is being measured in which panel; visual inspection does not correlate with the alleged result and there is a confusing discussion about 2 upregulated miRNA's from parent not consistent with initial paper reporting on 6 downregulated miRNA's.
Authors
Indeed, for the father's sperm, see the publication Ozkul et al., 2021, most of the miRNAs among the six identified in the blood were also downregulated, but two miRNAs, their reverse homolog (miR-499-3p and miR -361 - 5p) were upregulated in exaggerated proportions in the sperm of the father analyzed. We have changed the sentences in the text.
Reviewer 2
Are the control bars in these panels from mice not microinjected?
Authors
We have two types of control groups: non injected and microinjected with RNA from sperm (human control) .
Reviewer 2
Several behavioural tests were performed in 5 groups of animals, labelled Control, F0 SH, F0 ASH, F1 SH, F1 ASH (nomenclature as in ¶2 of this review). Although F0 and F1 offspring appear in the cartoon (Figure 2) summarizing the experiment, this is not discussed in the methods.
Authors
Thank you for your attention. For the method, we have added information for the transmission to F1 generation.
Reviewer 2
Again, there are bars in the panels of Figure 5 indicating significance, but no statistical analysis is presented, and the sample sizes of each of the animal groups for each of the behavioural tasks is not noted.
Authors
Added and corrected.
The F1 generation was evaluated only for behavioral changes. In Figure 3 (Figure5), data for the F1 generation is already given in the results. Sample size has been added to figure legend.
Reviewer 2
I have no concern with the photomicrographs presented as examples of histochemical analysis, but the same concerns about sample size and statistical analysis pertain here.
Authors
We have added.
Reviewer 2
Throughout the manuscript there are awkward sentences and phrases and both the text and the reference list require careful reading and correction of both typographical and other errors: example, reference 15 comprises 2 references, and reference 16 refers to 2 different citations.
Authors
We apologize for the errors and corrected.
Comments on the Quality of English Language
Reviewer 2
The investigators obviously write many papers in English, but this one was either rapidly constructed or they normally have a reviewer who picks up the many lapses and typographical errors. This is less an issue of English per se, and more an issue of poor reviewing of the manuscript for readability, typos, and errors of statement.
Round 2
Reviewer 2 Report
Comments and Suggestions for Authors
This manuscript is improved in a number of aspects, but some materials added to the text raise new questions. For example, the sentence on lines 84-88 mentions sperm from "mice treated with valproic acid and miRNA candidates" but I do not find any information regarding this in the methods or behavioral results. If the addition of some slides from a valproate model are to be included, please reference the source in the main text, not only in supplemental figures.
Going through the manuscript in the order of presentation (not the best approach but timely) I have the following questions or concerns:
line 112: was the control sperm tested for the target miRNA's?
line 116: Presumably the term miR*refers to mice born from ova injected with the control sperm RNA. Are these the same animals as those identified in most of the figures as "Control"?
Results: the first 3 paragraphs of section 2.1 are redundant with information previously presented in the introduction; if it is necessary to reiterate some of this it could be condensed
Also, the sentence on lines 262-264 is inappropriate: if possession of the 6 target miRNA's causes
ASD, why is the transmitting father not affected?
2.4 immunohistochemistry: how was staining intensity measured?
Figure 10: do the authors mean Cingulate cortex? if not, what is the "singular" cortex?
Overall, readers that are knowledgeable about the complexity of miRNA production and metabolism are going to find the discussion to be overly simplistic, despite the fact that the data is interesting. Some "instructions for RNA variations" such as sequence variation are unquestionably transmitted, but control of miRNA expression is much more complex than this.
It may also be relevant that the discussion includes only 2 citations, neither of them related to miRNA expression.
manuscript needs to be proofread for punctuation and spelling
Author Response
Reviewer2: This manuscript is improved in a number of aspects, but some materials added to the text raise new questions. For example, the sentence on lines 84-88 mentions sperm from "mice treated with valproic acid and miRNA candidates" but I do not find any information regarding this in the methods or behavioral results. If the addition of some slides from a valproate model are to be included, please reference the source in the main text, not only in supplemental figures.
Authors
Yes, thanks for your previous reviews and here we have clarified the sentences. Here, living two- week-old young males were treated with valproic acid and then sperm were collected two months later from adults. The data on valproic acid male are already reported in our previous article, here we report histological analysis with glial cells alteration.
Reviewer2: Going through the manuscript in the order of presentation (not the best approach but timely) I have the following questions or concerns:
line 112: was the control sperm tested for the target miRNA's?
Authors
RNA from sperm were tested for miRNA but not for targets.
Reviewer2: line 116: Presumably the term miR*refers to mice born from ova injected with the control sperm RNA.
Authors
miR* refers to animals born after microinjection of a given miRNA with a phenotype different from controls, such as behavioral changes here.
Reviewer2: Are these the same animals as those identified in most of the figures as "Control"?
Authors
No, on the one hand we have controls who do not receive an microinjection or who receive a microinjection of human sperm control RNA.
Reviewer2: Results: the first 3 paragraphs of section 2.1 are redundant with information previously presented in the introduction; if it is necessary to reiterate some of this it could be condensed
Authors
That's right, we shortened and changed the sentence.
Reviewer2: Also, the sentence on lines 262-264 is inappropriate: if possession of the 6 target miRNA's causes
Authors
We have not found the corresponding sentence, what you mean? please give more information.
Reviewer2: ASD, why is the transmitting father not affected?
Authors
Because miRNA levels are not as low as in the patient, his children have much lower blood levels of six miRNAs than their father. There is a decrease in the level of miRNAs in the blood of parents compared to controls, but it is higher than that of patients. This is already discussed in our article in Ozkul et al. We added a sentence.
Reviewer2: 2.4 immunohistochemistry: how was staining intensity measured?
Authors
JPEG images were imported into ImageJ/Fiji software to measure immunohistological staining for each protein. The threshold function was applied to separate the signal from the background and the average signal intensity was measured with the “measure” function.
Reviewer2: Figure 10: do the authors mean Cingulate cortex? if not, what is the "singular" cortex?
Authors
Yes, you are right, we have corrected apologize.
Reviewer2: Overall, readers that are knowledgeable about the complexity of miRNA production and metabolism are going to find the discussion to be overly simplistic, despite the fact that the data is interesting.
Authors
Yes, the system is very complex, we are trying to simplify the complexity of the system, that does not mean we have ignored it.
Reviewer2: Some "instructions for RNA variations" such as sequence variation are unquestionably transmitted, but control of miRNA expression is much more complex than this.
Authors
You are absolutely right, we used your expression.
Reviewer2: It may also be relevant that the discussion includes only 2 citations, neither of them related to miRNA expression.
Authors
We added, thanks for pointing that out.